# A Human Feedback Strategy for Photoresponsive Molecules in Drug Delivery: Utilizing GPT-2 and Time-Dependent Density Functional Theory Calculations

**DOI:** 10.3390/pharmaceutics16081014

**Published:** 2024-07-31

**Authors:** Junjie Hu, Peng Wu, Shiyi Wang, Binju Wang, Guang Yang

**Affiliations:** 1Faculty of Medicine, Imperial College London, London SW7 2AZ, UK; j.hu@imperial.ac.uk; 2School of Chemistry and Chemical Engineering, Ningxia University, Yinchuan 750014, China; 3Bioengineering Department and Imperial-X, Imperial College London, London W12 7SL, UK; s.wang22@imperial.ac.uk; 4College of Chemistry and Chemical Engineering, Xiamen University, Xiamen 361005, China; wangbinju2018@xmu.edu.cn; 5National Heart and Lung Institute, Imperial College London, London SW7 2AZ, UK; 6Cardiovascular Research Centre, Royal Brompton Hospital, London SW3 6NP, UK; 7School of Biomedical Engineering & Imaging Sciences, King’s College London, London WC2R 2LS, UK

**Keywords:** drug delivery, photoresponsive molecule, GPT, TDDFT, RLHF

## Abstract

Photoresponsive drug delivery stands as a pivotal frontier in smart drug administration, leveraging the non-invasive, stable, and finely tunable nature of light-triggered methodologies. The generative pre-trained transformer (GPT) has been employed to generate molecular structures. In our study, we harnessed GPT-2 on the QM7b dataset to refine a UV-GPT model with adapters, enabling the generation of molecules responsive to UV light excitation. Utilizing the Coulomb matrix as a molecular descriptor, we predicted the excitation wavelengths of these molecules. Furthermore, we validated the excited state properties through quantum chemical simulations. Based on the results of these calculations, we summarized some tips for chemical structures and integrated them into the alignment of large-scale language models within the reinforcement learning from human feedback (RLHF) framework. The synergy of these findings underscores the successful application of GPT technology in this critical domain.

## 1. Introduction

Smart drug delivery systems, which have gained significant attention in pharmaceutical research, enhance patient health by ensuring targeted therapeutic delivery [1]. In recent decades, artificial intelligence (AI) has demonstrated its capability to address complex challenges within pharmaceutical research, particularly in smart drug delivery, from the observable to the micro-/nanoscale [2,3,4]. These advanced drug delivery systems are designed to be self-regulating, responding to a range of stimuli associated with disease pathology [5,6]. Light stands out as an external actuation method for therapeutic applications due to its non-invasive characteristics, stability in biological settings, adjustable intensity, and unparalleled temporal and spatial precision [7,8,9,10,11,12].

It is widely used in smart delivery systems for various therapeutics, with different wavelengths ranging from ultraviolet (UV, 100–400 nm) to visible (400–750 nm) and near-infrared (NIR, 750–2000 nm) light eliciting different responses [13]. Short-wavelength light, or UV light, has enough energy to alter chemical bonds and configurations, such as breaking covalent bonds or changing cis–trans conformations, effectively triggering drug delivery mechanisms [14,15]. Owing to these advantages, UV light is frequently used as a stimulus in diverse research and applications [16,17].

It is expected that deep learning will also be used in the solutions of more problems in drug delivery, including the design of stimulus-responsive molecules [4,18]. The prediction of temporal dynamics in drug delivery has been accomplished through the application of convolutional neural networks and long short-term memory networks [19]. When deep learning is used to solve molecular design problems in chemistry and materials, the lack of specialized datasets often leads to task failure. Advancements in pre-training and fine-tuning methods for large language models (LLMs) have facilitated the creation of chemical molecules [20,21]. The potential utilization of the generative pre-trained transformer for designing light-responsive molecules is significant, and there is currently limited research in this specific domain.

Causal language models such as GPT, GPT-2, and GPT-3 are trained to calculate/predict the probability of the occurrence of several words given all preceding words, making them ideal for text generation [22,23,24]. After character-level RNNs and masked transformer language models were used to capture Simplified Molecular Input Line Entry System (SMILES) language patterns [21,25], Sanjar Adilov constructed a GPT-2-like language model to learn SMILES representations and transfer knowledge to downstream molecular generative tasks [26]. In subsequent studies, GPT has attracted the attention of more researchers [27,28]. Additionally, instruction tuning using human experience to enhance large models has proven to be a highly effective method for improving the quality of generated content. This approach includes the use of Direct Preference Optimization (DPO) algorithms [29], Contrastive Preference Optimization (CPO) algorithms [30], and others. Recently, researchers have also proposed Kahneman–Tversky Optimization (KTO) algorithms to simplify dataset preparation for this purpose [31]. Our integration of GPT-2 with these techniques for applications in light-responsive drug delivery represents a very promising and meaningful exploration.

The open-source dataset QM7b, provided by the Quantum Machine Project, encompasses a variety of physicochemical properties of molecules [32,33]. In order to ultimately achieve the generation of light-responsive drug delivery molecules, these data were used to fine-tune the pre-trained language model [33,34]. Among the physicochemical properties in the dataset are the excitation energies of the molecules. Our UV-GPT inherits the transformer structure from GPT-2 and utilizes the tokenizer from SMILES-GPT, facilitating the downstream generation of UV-responsive molecules. By integrating predictive modeling and TDDFT calculations, we discovered that the fine-tuning UV-GPT model generates molecules with UV photoresponsive properties. The molecules generated were positively influenced by the pre-trained SMILES-GPT, as evidenced by the statistics on drug-like properties and synthesizability. This evidence shows that our application of the combined GPT and TDDFT calculations in designing stimulus-responsive molecules holds practical value for drug delivery. Additionally, we explored various implementations of RLHF, utilizing structural knowledge generated by computational chemistry as human feedback to enhance the quality of the generated content. However, our current model does not fully integrate the chemists’ extensive theoretical knowledge of chemistry.

## 2. Materials and Methods


**Pre-trained language model and adapter tuning**


We trained our model for 6 epochs on the SMILES strings of the QM7b datasets [32,33]. We used AdamW for optimization and cosine annealing for learning-rate scheduling. The initial and final learning rates were set to $5\times 10^{-4}$ and $5\times 10^{-8}$, respectively. We kept the default Adam hyperparameters and optimized the batch size (128) and maximum sequence length (512).

Our SMILES tokenizer was pre-trained on SMILES-GPT. This transformer decoder replicates GPT-2 except that during tokenization, it used the character-level byte-pair encoding instead of byte-level encoding. We reserved 72 characters from the SMILES alphabet as an initial vocabulary and supplemented the vocabulary with up to 1000 of the most frequent merges. The model uses parameterized token and position embeddings, 8 attention heads, and 4 attention blocks. With the embedding/hidden dimension of 512, it has 13.4 M parameters.


**Prediction model for excitation energy**


The Coulomb matrix of the molecule was generated using DeepChem computation. The training set and test set were divided in an 80:20 ratio. Implementation of Support Vector Regression (SVR) with a quantum kernel was achieved with sklearn and qiskit, while SVR with alternative kernels was implemented solely with sklearn.


**Drug-likeness and evaluation metrics for generative molecules**


Drug-likeness is a consideration when evaluating the generative molecules for photoresponsive drug delivery. We utilized the quantitative estimate of drug-likeness (QED) as a metric, as introduced in [35]. The QED metric yields a numerical score ranging from 0 to 1, where elevated scores correspond to an increased probability of drug-likeness.

The Synthetic Accessibility Score (SAscore) [36], used to assess the ease of synthesizing drug-like molecules, rates molecules from 1 to 10 based on historical synthetic data and molecular complexity. Fragment contributions and a complexity penalty form the basis, derived from PubChem’s vast molecule database. Validation against expert chemists’ estimations showed strong agreement (r^2^ = 0.89). This method harnesses big data to streamline and enhance the synthesis evaluation process in molecular design.


**DFT and TDDFT simulation**


Density functional theory (DFT) calculations were performed on the molecules using ORCA 5.0.4 [37]. Optimization of molecules was performed at the PBE0/6−311G∗ level of theory [38]. Time-dependent density functional theory (TDDFT) calculations were performed at the PBE0/TZVP level [38,39]. The gas phase, water, and chlorobenzene solvents were modeled using the implicit solvent polarizable continuum model (PCM) [40,41] with Grimme’s D3 [42,43,44] dispersion corrections during optimization and TDDFT calculations.


**RLHF**


We utilized the transformer reinforcement learning package for reinforcement learning with human feedback. In the fine-tuning process, we applied Direct Preference Optimization (DPO), Contrastive Preference Optimization (CPO), and Kahneman–Tversky Optimization (KTO) with the chemical knowledge datasets.

The default sigmoid loss was used in DPO, where the beta factor was set at 0.1. Similarly, the loss type of CPO was sigmoid, and its beta factor was set at 0.1. The beta factor in the KTO loss was set at 0.1, with a higher value meaning less divergence from the initial policy. The desirable and undesirable losses of KTO are weighed by desirable weight and undesirable weight, respectively. Both of them were set at 1.0.

The preference dataset used in DPO and CPO is a dictionary object with the keys ‘prompt’, ‘chosen’, and ‘rejected’. The binary signal dataset used in KTO is a dictionary object with the keys ‘prompt’, ‘completion’, and ‘label’.

## 3. Results and Discussion

### 3.1. Generative Workflow for Photoresponsive Drug Delivery

Numerous studies have focused on utilizing AI-based databases to scale up, optimize, and accelerate the development of nanocarrier drug delivery systems that are safe, effective, and stable. Endogenous triggers like pH variations, hormone levels, enzymatic actions, overexpression of biomarkers, glucose, or redox gradients are intrinsic to the body’s disease state. These triggers can externally prompt or amplify drug release in affected areas [6]. UV light’s high energy can modify chemical bonds, facilitating drug delivery mechanisms. Its versatility makes UV a common stimulus in research and applications for drug delivery. To design more effective UV-responsive drug delivery molecules, we employed pre-trained language models, fine-tuned target molecule datasets, and used machine learning predictive models of molecular excitation energies.

To date, no studies have established a methodology for applying GPT technology to drug delivery molecules and validating its efficacy. Here, we opted for a transformer structure with adapter layers, specifically the GPT-2 model. Pre-trained on the PubChem dataset, this GPT-2 model aims to generate molecules with high drug-likeness and easy synthesis. A SMILES tokenizer was developed by Sanjar Adilov, based on atomic dictionaries and linked characters. The linked characters of molecular SMILES provide information about chemical bonds and molecular configurations. The SMILES tokenizer also incorporates Multi-Layer Perceptron (MLP) and attention mechanism networks. Our goal is to adapt a pre-trained GPT for generating effective photoresponsive molecules using adapter layers. Once this workflow is proven to work, we can integrate various pre-trained language models through Hugging Face. This workflow for generating stimulus-responsive molecules via a pre-trained language model is shown in Figure 1.

The QM7b dataset provides excitation energies for molecules, which were utilized as input for fine-tuning our UV-GPT and training our screening model. This open-source dataset also includes the Coulomb matrix and physicochemical properties of molecules. Prof. Alexandre Tkatchenko’s group shared the SMILES of molecules with us [32,33], which are crucial for our adapter training and serve as the foundation of our workflow. Our generative pre-trained transformer for UV light-responsive drug delivery (UVGPT) utilized training datasets containing molecules with excitation energies ranging from 4.13 to 12.41 eV. Understanding the differences in the properties of various molecules based on chemical bonds, atomic potentials, and molecular conformations is a direct manifestation of the quantitative structure–activity relationship (QSAR) of molecules. Experienced chemical researchers can design molecules with specific properties based on their intuition. Our UVGPT learns the QSAR of UV light-responsive molecules from training datasets.

### 3.2. Screening the Generative Molecules

After training our UVGPT on UV light-responsive molecules, a total of one thousand molecules were generated by the model. Among these, 443 possessed valid chemical structures processed with RDKit. Dylan M. Anstine and Olexandr Isayev [45] reviewed generative methods in chemical sciences and proposed evaluation metrics. The methods for calculating drug-likeness and SAscore, as outlined in their publications [45], were employed in our research. QED estimated the drug-likeness of molecules using a machine learning model trained on a dataset of drug-like compounds. Higher QED values suggest an increased likelihood of drug-likeness. Similarly, the Synthetic Accessibility Score calculation methods aimed to systematically assess the ease of synthesizing drug-like molecules, aiding in prioritizing compounds for molecular design.

In the following workflow, we evaluated the excitation energy, drug-likeness, and SAscore of these molecules, as illustrated in Figure 2. We utilized the Coulomb matrix of molecules as the molecular descriptor in our prediction model. Support Vector Regression (SVR) served as the predictive machine learning model. We methodically compared various sets of parameters for SVR, encompassing different kernels (rbf, sigmoid, and quantum), regularization parameters (from 0.01 to 80), and kernel coefficients (auto, scale, 0.8, 0.84, and 2.3). Based on the Mean Squared Error (MSE) of both the training and test sets, our predictive model selected the rbf kernel, a regularization parameter of 80, and a kernel coefficient of 0.84. Prior to utilizing SVR for predicting the excitation energies of generative molecules, we employed DeepChem to compute the Coulomb matrix of these generative molecules. The results are shown in Figure 2a.

When analyzing the density distribution plots of excitation energies depicted in Figure 2a, we observe a range spanning from 5 to 11 eV, with the majority concentrated between 8 and 9 eV. This distribution closely aligns with that of the training samples, indicating UVGPT’s success in learning the QSAR from molecular data and facilitating molecular design.

To refine the assessment of molecule excitation energies via precise TDDFT quantum chemical calculations, we conducted additional screening of the molecules. Given that drug delivery molecules, while not directly engaged in target binding at the lesion site, still exert direct effects on the human body, drug-like characteristics are equally crucial for the generated molecules. The parameter-tuned UVGPT inherited the pre-trained model’s performance on PubChem, and the distribution calculated using the QED method is illustrated in Figure 2b.

Based on the QED ranking, we identified the nine molecules with the highest degree of drug-like properties, and their corresponding SMILES representations and QED values are shown in Figure 3. These nine molecules exhibited QED values ranging from 0.528 to 0.57. Notably, the molecule with the SMILES notation OC1CC1OC(C)C demonstrated the highest degree of drug-like properties and was absent from the PubChem database. Additionally, the compound (1S,2S)-2-methylcyclopropan-1-ol in PubChem shared similarities with OC1CC1OC(C)C. Both findings further contribute to the evidence demonstrating the effectiveness of UVGPT.

Similarly, based on the density distribution results of the SAscore in Figure 2c, we identified eight recommended molecules after filtering out mediocre results. The SAscore values of these eight molecules ranged from 5.47 to 5.92. Notably, the molecule with the SMILES notation C=CC=NSN=C achieved an SAscore value of 5.537, indicating its status as a conjugated alkene. Additionally, the molecule with the SMILES notation CC(C(Cl)N)S=N also achieved an SAscore value of 5.537, classifying it as a cumulene. These are shown in Figure 4.

### 3.3. Quantum Chemical Calculations

To enhance our understanding and validate the outcomes generated by UVGPT, we conducted quantum chemical simulations to compute the physicochemical properties of the screened molecules, focusing primarily on their excitation energies. Additionally, we assessed the rationality and stability of the molecular structures within the chemical context.

To validate the excited state properties, vertical excitation by DFT calculations were performed. As shown in Table 1, nearly all (16/17) generated molecules contained heteroatoms, including oxygen, sulfur, chlorine, and especially nitrogen, suggesting potential for biological applications. Six molecules were saturated with a first excited energy greater than 6.199 eV (putative corresponding maximum absorption peak wavelength < 200 nm). Seven molecules exhibited a maximum absorption peak wavelength between 200 nm and 400 nm. The remaining five molecules were in the visible and near-infrared bands. Enhancements in the generalization of predictive models could contribute to increased efficiency in our smart workflow.

Moreover, as discussed earlier, the molecule with the SMILES notation OC1CC1OC(C)C is significant for drug application. Unfortunately, this molecule exhibits far-ultraviolet absorption and is unlikely to be used in drug delivery applications. Additionally, we did not find useful analogs in the databases. Therefore, we propose that modifying the isopropyl group by introducing double bonds (e.g., aldehyde, nitro, etc.) could decrease the excitation energy, potentially making it suitable for photoresponsive molecules with special drug applications.

It is widely recognized that conjugated alkenes are more stable than cumulenes. The cumulene molecule generated by UVGPT does not conform to an optimal structure within the bounds of established chemical knowledge. For example, the molecule with the SMILES notation CC1NC(C)=C=C1 (cumulene) lies several kcal/mol higher in Gibbs energy compared to its isomer. Addressing this issue requires incorporating additional chemists’ intuition and improving the quality of the datasets utilized.

Additionally, the conjugated alkene with the SMILES notation C=CC=NSN=C exhibits ultraviolet absorption properties. Its ability to undergo photochemical reactions upon UV light exposure renders it potentially useful in drug delivery applications.

### 3.4. Fine-Tuning with Human Feedback

By combining the results of our quantum chemical simulations shown in Table 1 with the molecular structures shown in Figure 3 and Figure 4, we have leveraged our chemical knowledge to establish three structural criteria for identifying molecules with longer excitation wavelengths:The molecule has a polyatomic ring structure.The atomic ring structure contains more than one unsaturated chemical bond.The ring structure includes non-carbon atoms, such as nitrogen (N), sulfur (S), oxygen (O), etc.

We filtered the molecular data according to these three criteria, resulting in 121 molecules that meet a weakened criterion and 251 molecules that do not meet the criteria at all. The weakened criterion means satisfying at least two of the three conditions.

As shown in Figure 5, the KTO instruction fine-tuning relies on a binary signal dataset labeled based on human experience. We labeled 121 molecules that meet the weakened criterion as ‘True’ and those that do not meet the criteria at all as ‘False’, resulting in a binary signal dataset of 372 entries. Additionally, the DPO and CPO algorithms rely on a preference dataset, which consists of 121 molecules for the recommended samples and another set for the rejected data. As noted by the proposer of the KTO algorithm, the training set for KTO is much easier to obtain under these conditions. Our findings in our application were consistent with this.

We further introduced a low-rank adapter based on the pre-trained GPT-2 to implement the fine-tuned model, aiming to make the generated molecular content more aligned with chemical experience and intuition. The structure of the fine-tuning model is shown in Figure 5, in which we updated the parameters of the adapter within the RLHF framework. The training parameters were primarily sourced from the relevant functional layers of the GPT-2 attention mechanism. Additionally, we fine-tuned the GPT-2-based model using KTO, DPO, and CPO trainers, respectively. We then counted the number of molecules in the generated content that satisfied the aforementioned weakened condition, with the results shown in Table 2. It was found that CPO produced a higher total number and proportion of valid molecular data. Although the dichotomous dataset for the KTO algorithm is more readily available, the quality of its generated content was significantly inferior to that produced by the other models in our scenario. Nonetheless, we recognize the potential of these three types of trainers, along with additional RLHF algorithms, for developing more effective language models for our task.

## 4. Future Work

Previous studies have highlighted the limitations of UV-responsive materials for in vivo applications, citing concerns like potential cellular photodamage and inadequate penetration [10]. In subsequent investigations, researchers have turned their attention to visible and near-infrared light-responsive molecular materials [11,46]. However, the development of molecules utilizing LLM modeling for this purpose remains hindered by the absence of high-quality datasets. We aim to tackle this challenge to enhance the applicability and credibility of our workflow for medical clinical applications.

The remarkable success of the generative pre-trained transformer in various applications has garnered significant attention from both academia and industry. Communities like AdapterHub and Hugging Face have amassed numerous open-source pre-trained transformer structures, facilitating the design of light-responsive molecules for drug delivery systems. This diversity of molecular generation tools and content enhances the applicability of these technologies. Moreover, these advancements can seamlessly integrate into the workflows discussed in this paper.

While our current models are proficient in inheriting properties from training datasets, they have yet to reach a level of innovative insight that could rival human chemists. There is a need to further explore methods for refining the model parameters using additional knowledge and more effective techniques. Reinforcement learning from human feedback (RLHF) methods provide an exciting opportunity to incorporate more theoretical chemical knowledge into generating high-quality molecular content. To achieve this goal, we need to invest further efforts into designing the entire algorithmic framework of the generative pre-trained transformer (GPT) from scratch.

## Figures and Tables

**Figure 1 pharmaceutics-16-01014-f001:**
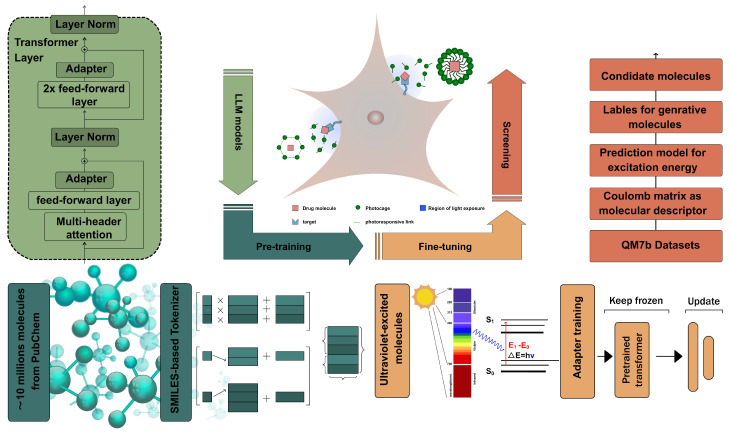
The workflow for generating UV light-responsive molecules. This workflow includes an LLM model based on GPT-2, a SMILES tokenizer, pre-training on PubChem datasets, fine-tuning on UV molecule datasets, and a screening model incorporating the Coulomb matrix descriptor. Molecules excited by ultraviolet light have the potential to become stimuli-responsive materials in drug delivery systems. The dark-green areas represent the pre-trained transformer. The indigo areas represent the pre-training of GPT-2 combined with the SMILES tokenizer on the PubChem dataset. The orange-yellow areas indicate that a new adapter was fine-tuned using the ultraviolet light-excited molecule dataset on the pre-trained GPT-2. The orange-red areas show the prediction model, trained on the QM7b dataset and Coulomb matrix features, which predicts the properties of molecules generated by the fine-tuned GPT-2.

**Figure 2 pharmaceutics-16-01014-f002:**
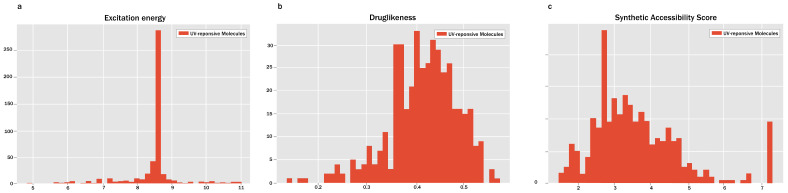
The excitation energy (**a**), drug-likeness (**b**), and SAscore (**c**) of generative molecules. These molecules were generated by GPT-2, fine-tuned on the ultraviolet light-excited molecule dataset. The excitation energy data in (**a**) are from predictions made by an SVM model. (**b**) Shows the distribution of the drug-likeness scores of the molecules, obtained from DeepChem’s calculation of the quantitative estimate of drug-likeness (QED) values. (**c**) presents the synthetic accessibility scores of the molecules, calculated using RDKit, with higher values indicating that the molecules are easier to synthesize.

**Figure 3 pharmaceutics-16-01014-f003:**
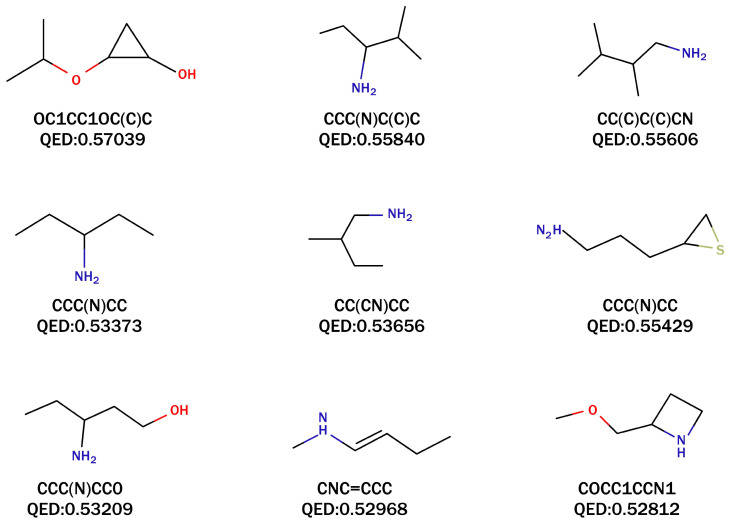
The QED values and SMILES representations of nine selected molecules generated by UVGPT.

**Figure 4 pharmaceutics-16-01014-f004:**
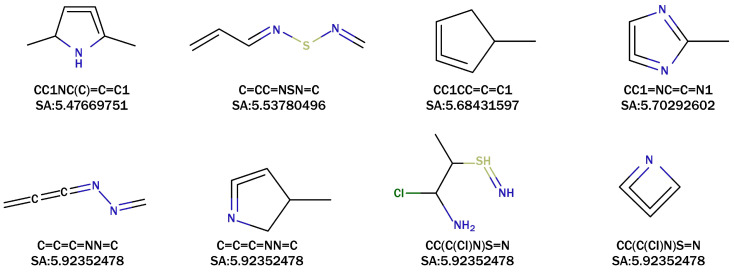
The SAscore values and SMILES representations of eight selected molecules generated by UVGPT.

**Figure 5 pharmaceutics-16-01014-f005:**
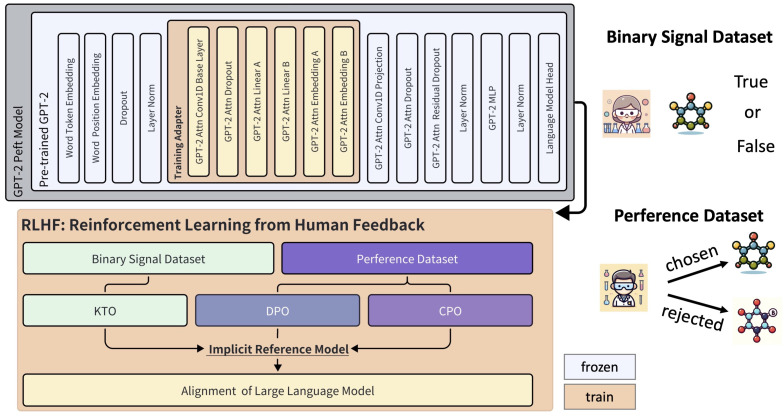
Instruction tuning with chemical datasets. The details of the pre-trained GPT-2 model are provided, including the implementation process of RLHF, which involves KTO, DPO, and CPO. The characteristics of the binary signal dataset and the preference dataset are also shown. In the training and fine-tuning of the GPT-2 Peft model, the parameters in the orange-yellow area were used for training, while the parameters of the other layers retained their pre-trained values. The KTO algorithm used the binary signal dataset, while both DPO and CPO used the preference dataset. Refer to the Methods section for descriptions of the binary signal and preference datasets. Labels in the binary signal dataset are assigned by experts to the molecules as either True or False. In the preference dataset, experts select recommended molecules and identify unreasonable molecules as restricted.

**Table 1 pharmaceutics-16-01014-t001:** Molecular structures and first excitation energies converted to wavelengths in the gas phase, water, and organic solvents.

Molecular SMILES	First Excitation Energy Converted to Wavelengths		
	* **Gas Phase** *	* **Water** *	* **Organic Solvents** *
CCC(N)CC	190	178	181
OC1CC1OC(C)C	191	177	180
CCC(N)C(C)C	193	182	185
CC(CN)C(C)	193	181	184
CC(C)C(C)CN	197	187	190
CCC(N)CCO	199	181	185
COCC1CCN1	201	189	192
CNC=CCC	231	219	222
CC(C(Cl)N)S=N	226	305	218
C1=C=C=N1	235	235	235
NCCCC1SC1	251	246	247
C=CC=NSN=C	286	288	291
C=C=C=NN=C	468	445	451
CC1CC=C=C1	492	696	605
CC1C=C=NC1	631	644	648
CC1NC(C)=C=C1	733	481	529
CC1=NC=C=N1	749	736	739

**Table 2 pharmaceutics-16-01014-t002:** Analysis of the generated content from the RLHF fine-tuned model.

RLHF Trainer	Number of Molecules Satisfying Different Conditions	
	* **At Least Two Judgments** *	* **Ratio of Two Judgments** *
KTO	16	21.05%
DPO	124	35.13%
CPO	131	43.96%

## Data Availability

Data files and Codes could be found via this link: https://github.com/jhu22/Pharmaceutics2024.

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
