# Peer review of "A Human Feedback Strategy for Photoresponsive Molecules in Drug Delivery: Utilizing GPT-2 and Time-Dependent Density Functional Theory Calculations"

_pharmaceutics, 2024, doi:10.3390/pharmaceutics16081014_

Round 1
Reviewer 1 Report
Comments and Suggestions for Authors
The manuscript presents a novel approach by combining a GPT model with TDDFT calculations to design UV-responsive molecules for drug delivery. This integration of AI with computational chemistry simulations is a relatively new concept and holds promise for accelerating the discovery of photoresponsive drug delivery systems. Furthermore, the authors explore the use of RLHF to refine the model, which is also innovative.
However, the latter should address the following issues to enhance the manuscript's scientific rigor:
-The laauthors acknowledge that their model does not fully integrate the extensive theoretical knowledge of chemists. This could be a significant limitation, as chemical intuition and expertise are often crucial in molecular design.
-The reliance on the QM7b dataset for training and validation could be further discussed: QM7b may not fully represent the diversity of molecules relevant to drug delivery. A more specialized dataset could improve the model's performance.
-The validation of the generated molecules was primarily based on computational simulations (TDDFT). While these simulations are valuable, experimental validation would significantly strengthen the study's conclusions.
Comments on the Quality of English Language
there are several minor language and English issues :
"the Generative Pre-trained Transformer (GPT) has been employed for generating molecular structures" (Introduction): "to generate" instead of "for generating"
"these evidences indicate that our application of the combined GPT and TDDFT calculations in designing stimulus-responsive molecules holds practical value for drug delivery" (Introduction): "evidences" should be singular
"the forecasting of temporal dynamics in drug delivery has been accomplished through the application of convolutional neural networks and long short-term memory networks [19]" (Introduction): "forecasting" could be replaced with "prediction".
"benefiting from the open-source QM7b datasets provided by Quantum-Machine Project [34,35], our photoresponsive molecular design for drug delivery has gained important datasets [36,37]" (Results and Discussion) : This sentence is unclear and should be rephrased.
"to date, studies haven't shown a clear path for applying GPT technology to drug delivery molecules and validating its efficacy" (Results and Discussion): "clear path" could be replaced with "established methodology"
"The linked characters of a molecular SMILES give out the information about the chemical bond and molecular configuration" (Results and Discussion): "gives" instead of "give"
Etc...
Author Response
[The revised paper can be viewed in the attachment]
Comments: -The laauthors acknowledge that their model does not fully integrate the extensive theoretical knowledge of chemists. This could be a significant limitation, as chemical intuition and expertise are often crucial in molecular design.
Reply: We strongly agree with this comment from the reviewers. Regarding to the limitations mentioned above, we have explained them in the following aspects:
Firstly, theories such as Valence Bond Theory, Molecular Orbital Theory, and Density Functional Theory provide a rich, multi-level theoretical foundation in the field of chemistry. Describing molecular structures remains an important and indispensable part. Our work primarily combines existing techniques to explore the potential of the GPT-2 model in molecular design, starting from molecular structure knowledge. We have also introduced DFT methods into our workflow to address the current limitations of the GPT-2 model in handling advanced chemical knowledge.
Secondly, our evaluation of this work is a compromise between expected goals and achievable methods. Even so, the obtained results show that integrating molecular structure knowledge related to molecular properties into GPT-2 using the RLHF method is effective. The limitations we pointed out mainly refer to the fact that the SMILES tokenizer we used is insufficient to support the learning of more extensive chemical knowledge. We have also supplemented the main text to emphasize this point.
Finally, our work in this paper does not utilize multimodal methods. Compared to GPT-2, more advanced large language models (LLMs) are considered to have the potential to handle more complex and systematic chemical knowledge in terms of content and structure. However, we are still unable to directly apply the multimodal large models, which have been successfully used for converting between images, text, and video, to the chemistry-related field. A multimodal large model capable of fully understanding multi-level chemical theories and performing molecular design as needed is the result we anticipate. Achieving this will require us to solve more technical problems and accumulate high-quality specialized data.
Comments: - The reliance on the QM7b dataset for training and validation could be further discussed: QM7b may not fully represent the diversity of molecules relevant to drug delivery. A more specialized dataset could improve the model's performance.
Reply: As the reviewers pointed out, we lack specialized, high-quality data on light-responsive drug delivery molecules. In this work, we primarily considered the excitation energy properties of light-responsive molecules. The QM7b dataset provides crucial references for the excitation energy properties and other physicochemical properties of molecules. It is one of the best open-source datasets currently available to us. In future work, we will also attempt to perform TDDFT calculations of excitation energies for molecules obtained from the literature and other sources to expand the dataset, though this will require some time to accumulate.
Additionally, the molecular data obtained from QM7b were used as the training set for fine-tuning the GPT-2 model. To enhance the drug-related properties of the molecules generated by GPT-2, we pre-trained the model using PubChem. The reviewers' suggestion to make the model more reflective of drug delivery molecules is, in our opinion, very wise and correct. The reason we did not fully implement the reviewers' suggestion is that drug delivery remains a dynamic and emerging research field. Moreover, the methods of stimulus-responsive drug delivery are diverse, which makes the scale and acquisition of light-responsive drug delivery molecules challenging.
Our strategy in this work was to pre-train the GPT-2 model on PubChem to meet the requirements of synthesizability and drug-likeness for drug delivery molecules. Although this does not fully represent the correlations that high-quality drug delivery molecules should meet, it does partially compensate for the limitations of the QM7b excitation energy dataset. In future work, we plan to start from the chemical mechanisms of light-responsive drug delivery molecules, combining quantum chemistry and reported results in the literature, to extract chemical knowledge and experience for the model. This will enhance the relevance of the molecules generated by the GPT-2 model to drug delivery.
Comments: - The validation of the generated molecules was primarily based on computational simulations (TDDFT). While these simulations are valuable, experimental validation would significantly strengthen the study's conclusion.
Reply: Mature photochemical and photophysical experimental conditions ensure the accurate measurement of molecular crystals and solvent properties. We fully agree with the reviewers that incorporating these data would significantly enhance the quality of our work. Unfortunately, synthesizing the molecules generated by the GPT-2 model requires a longer time frame. We believe that before proceeding with synthesis and property measurements, it would be wiser and more economical to enrich the GPT-2 and other LLMs with more comprehensive chemical knowledge.
Comments: Comments on the Quality of English Language
there are several minor language and English issues :
"the Generative Pre-trained Transformer (GPT) has been employed for generating molecular structures" (Introduction): "to generate" instead of "for generating"
"these evidences indicate that our application of the combined GPT and TDDFT calculations in designing stimulus-responsive molecules holds practical value for drug delivery" (Introduction): "evidences" should be singular
"the forecasting of temporal dynamics in drug delivery has been accomplished through the application of convolutional neural networks and long short-term memory networks [19]" (Introduction): "forecasting" could be replaced with "prediction".
"benefiting from the open-source QM7b datasets provided by Quantum-Machine Project [34,35], our photoresponsive molecular design for drug delivery has gained important datasets [36,37]" (Results and Discussion) : This sentence is unclear and should be rephrased.
"to date, studies haven't shown a clear path for applying GPT technology to drug delivery molecules and validating its efficacy" (Results and Discussion): "clear path" could be replaced with "established methodology"
"The linked characters of a molecular SMILES give out the information about the chemical bond and molecular configuration" (Results and Discussion): "gives" instead of "give"
Etc...
Reply: Based on the reviewers' comments and our rechecking, we have made the following changes. All changes have been highlighted as required.
- "the Generative Pre-trained Transformer (GPT) has been employed for generating molecular structures" (Introduction): " for generating " replaced with "to generate"
- "these evidences indicate that our application of the combined GPT and TDDFT calculations in designing stimulus-responsive molecules holds practical value for drug delivery" (Introduction): "evidences" replaced with “evidence”
- "the forecasting of temporal dynamics in drug delivery has been accomplished through the application of convolutional neural networks and long short-term memory networks [19]" (Introduction): "forecasting" replaced with "prediction".
- "benefiting from the open-source QM7b datasets provided by Quantum-Machine Project [34,35], our photoresponsive molecular design for drug delivery has gained important datasets [36,37]" (Results and Discussion): replaced with “The open-source dataset QM7b, provided by Quantum-Machine Project, encompasses a variety of physicochemical properties of molecules. In order to ultimately achieve the generation of light-responsive drug delivery molecules, this data was used to fine-tune the pre-trained language model”.
- "to date, studies haven't shown a clear path for applying GPT technology to drug delivery molecules and validating its efficacy" (Results and Discussion): "clear path" replaced with "established methodology"
- "The linked characters of a molecular SMILES give out the information about the chemical bond and molecular configuration" (Results and Discussion): "give" replaced with “gives”
- "The SMILES-Tokenizer is also deal with" replaced with "The SMILES-Tokenizer also deals with."
- "Once this workflow is proved to work" replaced with "Once this workflow is proven to work."
- "This workflow to generate stimulus-responsive molecules via pre-training language model could be found in Fig 1." replaced with "This workflow for generating stimulus-responsive molecules via a pre-trained language model can be found in Fig 1."
- "utilized as input" replaced with be "which are utilized as input."
- "crucial for our adapter training and serving as the foundation of our workflow" replaced with "which are crucial for our adapter training and serve as the foundation of our workflow."
- ”Quantitative structure–activity relationship (QSAR) of molecules stems from the direct influence of chemical bonding, atomic potentials, and molecular conformation on their properties.” replaced with " Understanding the differences in the properties of various molecules based on chemical bonds, atomic potentials, and molecular conformations is a direct manifestation of the quantitative structure-activity relationship (QSAR) of molecules."
- "RDkit" replaced with "RDKit"
- "Higher QED values suggest increased likelihood of drug-likeness" replaced with "Higher QED values suggest an increased likelihood of drug-likeness."; "Synthetic Accessibility Score calculation methods aimed" replaced with "the Synthetic Accessibility Score calculation methods aimed"
- "rbf, sigmoid and quantum" replaced with "rbf, sigmoid, and quantum"; "regularization parameter (from 0.01 to 80)" replaced with "the regularization parameter (from 0.01 to 80)"; "kernel coefficient (auto, scale, 0.8, 0.84, and 2.3)" replaced with "the kernel coefficient (auto, scale, 0.8, 0.84, and 2.3)"; "Before we used our SVR of excitation energies to predict the value of generative molecules, the deepchem was used to calculate the Coulomb matrix of generative modelcules." has been removed; "the deepchem" replaced with "DeepChem"; "The results could be found in Figure 2a" replaced with "The results can be found in Figure 2a"
- "The parameter-tuned UVGPT inherited the pre-trained model's performance on PubChem, with the distribution calculated using the QED method illustrated in Figure 2b." replaced with "The parameter-tuned UVGPT inherited the pre-trained model's performance on PubChem, and the distribution calculated using the QED method is illustrated in Figure 2b."
- "molecules with SMILES of OC1CC1OC(C)C" replaced with "molecules with the SMILES OC1CC1OC(C)C"
- "Both of the above further contribute to the evidence supporting the effectiveness of UVGPT." replaced with "Both findings further contribute to the evidence supporting the effectiveness of UVGPT."
- "the molecule with the SMILES representation of C=CC=NSN=C" replaced with "the molecule with the SMILES representation C=CC=NSN=C"
- "Additionally, the molecule with the SMILES representation of CC(C(Cl)N)S=N" replaced with "Additionally, the molecule with the SMILES representation CC(C(Cl)N)S=N"; "classifying it as cumulene" replaced with "classifying it as a cumulene."
- "These have been shown in Figure 4" replaced with "These are shown in Figure 4."
- "excitation energy properties" replaced with "excitation energies"
- "the vertical excitation by DFT calculations were performed" replaced with "vertical excitations by DFT calculations were performed"
- "generated molecules contains heteroatom" replaced with "generated molecules contain heteroatoms; "Sulfer" replaced with "sulfur"; "atoms suggesting the prospects for biological applications" replaced with "atoms, suggesting prospects for biological applications"
- "The remaining 5 molecules" replaced with "The remaining five molecules"
- "the molecule, with the SMILES of OC1CC1OC(C)C" replaced with "the molecule with the SMILES OC1CC1OC(C)C"
- "And we did not get useful analogues in databases" replaced with "Additionally, we did not find useful analogues in databases"
- "Herein, we propose the modification of the isopropyl by intruding double bonds (aldehyde, nitro etc.) can decrease the excitation energy" replaced with "Herein, we propose that modifying the isopropyl by introducing double bonds (e.g., aldehyde, nitro, etc.) can decrease the excitation energy"
- "the molecule with the SMILES of CC1NC(C)=C=C1" should be "the molecule with the SMILES CC1NC(C)=C=C1"; "lies kcal/mol higher than the isomer of in Gibbs energy" should be "lies kcal/mol higher than the isomer in Gibbs energy"
- "with the SMILES of C=CC=NSN=C" should be "with the SMILES C=CC=NSN=C"
- "establish three structural judgment criteria for obtaining molecules with longer excitation wavelengths" replaced with "establish three structural criteria for identifying molecules with longer excitation wavelengths"
- "binary signal dataset labeled based on human experience" should be "a binary signal dataset labeled based on human experience"
- "What we discovered in our application was still consistent." replace with "Our findings in our application were consistent with this."
- "where we updated the parameters of the adapter within the RLHF framework" should be rephrased for clarity: "in which we updated the parameters of the adapter within the RLHF framework."
- "Additionally, we fine-tuned the GPT-2-based model using KTO, DPO, and CPO trainers respectively." should be: "Additionally, we fine-tuned the GPT-2-based model using KTO, DPO, and CPO trainers, respectively."
- "light-responsive molecules in drug delivery systems" should be "light-responsive molecules for drug delivery systems"
- "This diversity in molecule generation tools and content enhances the applicability of these technologies." replaced with "The diversity of molecule generation tools and content enhances the applicability of these technologies."
- "Reinforcement Learning from Human Feedback (RLHF) methods provide an exciting opportunity to incorporate more theoretical chemical knowledge into the generation of high-quality molecular content." replaced with "Reinforcement Learning from Human Feedback (RLHF) methods provide an exciting opportunity to incorporate more theoretical chemical knowledge into generating high-quality molecular content."
- "designing the entire algorithmic framework of the generative pre-trained transformer (GPT) from scratch" replaced with "designing the entire algorithmic framework of the Generative Pre-trained Transformer (GPT) from scratch."

Reviewer 2 Report
Comments and Suggestions for Authors
Dear Authors,
As a reviewer, here are some comments and suggestions to improve the manuscript:
Please provide more details on the implementation of the KTO, DPO, and CPO algorithms for RLHF, including the specific parameters, settings, and datasets used for human feedback preparation.
What were the size and composition of the QM7b dataset used for pre-training and fine-tuning the UV-GPT model, and what potential limitations or biases might this dataset have?
Why were the QED and SAscore metrics chosen for screening the generative molecules, and what potential limitations or alternatives exist for these metrics?
Could you comment on the computational cost and scalability of the quantum chemical calculations for larger molecular datasets?
You propose incorporating more theoretical chemical knowledge into the RLHF framework. Could you provide specific examples or strategies for doing so, and discuss the potential challenges involved?
You mention the potential integration of various pre-trained Transformer structures from platforms like AdapterHub and Hugging Face. What challenges and opportunities might be associated with this integration, and how could it affect the performance and generalizability of your approach?
Could you provide a more thorough comparison of your approach with existing methods or benchmarks for molecular design in drug delivery applications, if available?
Have you considered the potential ethical implications and concerns associated with the use of AI and generative models in drug design and delivery, and how these could be addressed in future research?
Carefully proofread to fix minor typos, grammatical errors. Make sure abbreviations are defined at first use.
Author Response
[The revised paper can be viewed in the attachment]
Comments: Please provide more details on the implementation of the KTO, DPO, and CPO algorithms for RLHF, including the specific parameters, settings, and datasets used for human feedback preparation.
Reply: We have supplemented these details with new additions to the methods subsection on RLHF, as suggested by the reviewers, including the specific parameters, settings, and datasets used for human feedback preparation.
‘’’
The default sigmoid loss was used in DPO, where the beta factor was set at 0.1. Similarly, the loss type of CPO was sigmoid, and its beta factor was set at 0.1. The beta factor in KTO loss was set at 0.1, with a higher value meaning less divergence from the initial policy. The desirable losses and undesirable losses of KTO are weighed by desirable weight and undesirable weight, respectively. Both of them were set at 1.0.
The preference dataset, used in DPO and CPO, is a dictionary object with the keys of 'prompt,' 'chosen,' and 'rejected.' The binary signal dataset, used in KTO, is a dictionary object with the keys of 'prompt,' 'completion,' and 'label.'
‘’’
Comments: What were the size and composition of the QM7b dataset used for pre-training and fine-tuning the UV-GPT model, and what potential limitations or biases might this dataset have?
Reply: The limitations mentioned by the reviewers are also one of the challenges in conducting deep learning work in the field of drug delivery. The pre-trained GPT model partially mitigates the lack of specialized high-quality data and compensates for this using other methods. This is highlighted in our workflow in Figure 1. Additionally, RLHF is an effective approach for continually integrating more drug delivery knowledge. The KTO algorithm's binary dataset also offers significant advantages in preparing the training set.
Additionally, QM7b contains the physicochemical properties of molecules, which have a weak correlation with drug applications. We address this issue by pre-training with the PubChem dataset. An ideal approach would be to obtain a high-quality, large-scale dataset of light-responsive drug delivery from existing research, but this requires more time and financial investment. For physicochemical properties such as excitation energies of molecules, QM7b is a widely used and peer-recognized dataset for similar model testing
.
Comments: Why were the QED and SAscore metrics chosen for screening the generative molecules, and what potential limitations or alternatives exist for these metrics?
Reply: Considering the need for experimental determination of the physicochemical properties of molecules, the SAscore is a crucial metric. The application of light-responsive drug delivery molecules imposes certain requirements on molecular weight, toxicity, and metabolism. Here, we comprehensively considered the QED drug-likeness index for screening. D. M. Anstine et al. systematically reviewed the evaluation methods for molecular generation models in their work on “Generative Models as an Emerging Paradigm in the chemical Sciences, J. Am. Chem. Soc. 2023, 145, 16, 8746-8750”, which serves as a suitable reference for selecting alternative metrics.
Comments: Could you comment on the computational cost and scalability of the quantum chemical calculations for larger molecular datasets?
Reply: We thank the reviewer for this constructive advice. Cost is indeed an important issue for the quantum calculations especially for large scale calculations. Generally, DFT calculation cost scales cubically with the number of atoms in one molecule, and nearly linearly with the size of dataset if the molecule are of similar size. In the QM7b dataset, a typical molecule (e.g. C6H12O2) requires about 25,200 s for the optimization and corresponding TD-DFT calculations at level of PBE0/TZVP//6-311G* by a single processer of EPYC 7H12 server. Therefore, a dataset with 1000 molecules can take several days to complete quantum calculations.
Comments: You propose incorporating more theoretical chemical knowledge into the RLHF framework. Could you provide specific examples or strategies for doing so, and discuss the potential challenges involved?
Reply: The excitation energy of molecules is related to advanced theoretical chemistry knowledge. We also attempted to have RLHF directly learn molecules with higher or lower excitation energies. After preparing the training set and analyzing the results, we found that this attempt was unsuccessful in the current process. This is because the learning process essentially aims to establish a relationship between the SMILES sequences representing the atomic and chemical bond composition of molecules and their excitation energy data. Theoretically, with sufficient high-quality data, the GPT-2 model could achieve this. However, we are unable to estimate the scale and quality of data required to reach this goal.
We speculate that by designing a tokenizer that establishes more connections between SMILES sequences and excitation energies, as well as ensuring mutual translatability with molecular structures, the training effectiveness could be improved to some extent. Currently, we are exploring feasible and effective approaches to achieve this.
Comments: You mention the potential integration of various pre-trained Transformer structures from platforms like AdapterHub and Hugging Face. What challenges and opportunities might be associated with this integration, and how could it affect the performance and generalizability of your approach?
Reply: In the latest advancements of LLMs, improving the training performance of LLMs is an important goal. This is achieved by enhancing the computational efficiency of the attention mechanism. Thanks to model encapsulation, when applying these advancements, we mainly need to understand the syntax mentioned in the Hugging Face documentation. Additionally, fine-tuning related to model alignment in LLMs has been found to aid in specific domain applications. Computer scientists have provided us with numerous efficient tools for handling chemical knowledge. By combining theoretical chemistry knowledge with new tools, such as SchNet, we ideally believe that this has a positive impact on addressing current engineering and technical challenges.
Comments: Could you provide a more thorough comparison of your approach with existing methods or benchmarks for molecular design in drug delivery applications, if available?
Reply: Our use of GPT-2 for molecular generation does not significantly differ from previous molecular generation work. We recognize that important experimental and theoretical computational work in light-responsive drug delivery involves discussing reaction mechanisms and validating the efficacy of molecular applications. This field lacks a materials information database similar to the Materials Project, which limits the associated deep learning work. Our work attempts to enhance the effectiveness in specific application scenarios by incorporating direct chemical knowledge under these constraints. Therefore, it is challenging to provide benchmarks for comparison with different drug delivery deep learning efforts.
However, when we subsequently use a self-designed model to provide synthetic pathways for molecules, we will be able to offer very standardized benchmarks for comparison.
Comments: Have you considered the potential ethical implications and concerns associated with the use of AI and generative models in drug design and delivery, and how these could be addressed in future research?
Reply: In AI models for drug design and drug delivery, prediction models are also an important component. A key challenge for prediction models is to eliminate systematic random errors. By developing more interpretable prediction models, we aim to ensure the consistency between the predicted and actual values for different molecules. Distortion and incomparability of prediction data can introduce significant risks. We have noted that process tomography mentioned in quantum machine learning is a potential approach, and work like FermiNet also provides feasible ideas. These are key references for developing predictive models for molecules.
Similarly, LLMs perform molecular generation tasks through pre-training and fine-tuning datasets, and as model capabilities continue to improve, this will significantly enhance the efficiency of related industries. Throughout the lifecycle of the industry, AI models, including GPT, can be considered as one of the links and nodes. Reasonable allocation of rights is an effective way to prevent and mitigate ethical issues. Thus, AI models cannot be evaluated in isolation but should be reasonably designed within a more complete industrial chain.
Comments: Carefully proofread to fix minor typos, grammatical errors. Make sure abbreviations are defined at first use.
Reply: Based on the reviewers' comments and our rechecking, we have made the following changes. All changes have been highlighted as required. We will also edit all abbreviations in MPDI format.
- "the Generative Pre-trained Transformer (GPT) has been employed for generating molecular structures" (Introduction): " for generating " replaced with "to generate"
- "these evidences indicate that our application of the combined GPT and TDDFT calculations in designing stimulus-responsive molecules holds practical value for drug delivery" (Introduction): "evidences" replaced with “evidence”
- "the forecasting of temporal dynamics in drug delivery has been accomplished through the application of convolutional neural networks and long short-term memory networks [19]" (Introduction): "forecasting" replaced with "prediction".
- "benefiting from the open-source QM7b datasets provided by Quantum-Machine Project [34,35], our photoresponsive molecular design for drug delivery has gained important datasets [36,37]" (Results and Discussion): replaced with “The open-source dataset QM7b, provided by Quantum-Machine Project, encompasses a variety of physicochemical properties of molecules. In order to ultimately achieve the generation of light-responsive drug delivery molecules, this data was used to fine-tune the pre-trained language model”.
- "to date, studies haven't shown a clear path for applying GPT technology to drug delivery molecules and validating its efficacy" (Results and Discussion): "clear path" replaced with "established methodology"
- "The linked characters of a molecular SMILES give out the information about the chemical bond and molecular configuration" (Results and Discussion): "give" replaced with “gives”
- "The SMILES-Tokenizer is also deal with" replaced with "The SMILES-Tokenizer also deals with."
- "Once this workflow is proved to work" replaced with "Once this workflow is proven to work."
- "This workflow to generate stimulus-responsive molecules via pre-training language model could be found in Fig 1." replaced with "This workflow for generating stimulus-responsive molecules via a pre-trained language model can be found in Fig 1."
- "utilized as input" replaced with be "which are utilized as input."
- "crucial for our adapter training and serving as the foundation of our workflow" replaced with "which are crucial for our adapter training and serve as the foundation of our workflow."
- ”Quantitative structure–activity relationship (QSAR) of molecules stems from the direct influence of chemical bonding, atomic potentials, and molecular conformation on their properties.” replaced with " Understanding the differences in the properties of various molecules based on chemical bonds, atomic potentials, and molecular conformations is a direct manifestation of the quantitative structure-activity relationship (QSAR) of molecules."
- "RDkit" replaced with "RDKit"
- "Higher QED values suggest increased likelihood of drug-likeness" replaced with "Higher QED values suggest an increased likelihood of drug-likeness."; "Synthetic Accessibility Score calculation methods aimed" replaced with "the Synthetic Accessibility Score calculation methods aimed"
- "rbf, sigmoid and quantum" replaced with "rbf, sigmoid, and quantum"; "regularization parameter (from 0.01 to 80)" replaced with "the regularization parameter (from 0.01 to 80)"; "kernel coefficient (auto, scale, 0.8, 0.84, and 2.3)" replaced with "the kernel coefficient (auto, scale, 0.8, 0.84, and 2.3)"; "Before we used our SVR of excitation energies to predict the value of generative molecules, the deepchem was used to calculate the Coulomb matrix of generative modelcules." has been removed; "the deepchem" replaced with "DeepChem"; "The results could be found in Figure 2a" replaced with "The results can be found in Figure 2a"
- "The parameter-tuned UVGPT inherited the pre-trained model's performance on PubChem, with the distribution calculated using the QED method illustrated in Figure 2b." replaced with "The parameter-tuned UVGPT inherited the pre-trained model's performance on PubChem, and the distribution calculated using the QED method is illustrated in Figure 2b."
- "molecules with SMILES of OC1CC1OC(C)C" replaced with "molecules with the SMILES OC1CC1OC(C)C"
- "Both of the above further contribute to the evidence supporting the effectiveness of UVGPT." replaced with "Both findings further contribute to the evidence supporting the effectiveness of UVGPT."
- "the molecule with the SMILES representation of C=CC=NSN=C" replaced with "the molecule with the SMILES representation C=CC=NSN=C"
- "Additionally, the molecule with the SMILES representation of CC(C(Cl)N)S=N" replaced with "Additionally, the molecule with the SMILES representation CC(C(Cl)N)S=N"; "classifying it as cumulene" replaced with "classifying it as a cumulene."
- "These have been shown in Figure 4" replaced with "These are shown in Figure 4."
- "excitation energy properties" replaced with "excitation energies"
- "the vertical excitation by DFT calculations were performed" replaced with "vertical excitations by DFT calculations were performed"
- "generated molecules contains heteroatom" replaced with "generated molecules contain heteroatoms; "Sulfer" replaced with "sulfur"; "atoms suggesting the prospects for biological applications" replaced with "atoms, suggesting prospects for biological applications"
- "The remaining 5 molecules" replaced with "The remaining five molecules"
- "the molecule, with the SMILES of OC1CC1OC(C)C" replaced with "the molecule with the SMILES OC1CC1OC(C)C"
- "And we did not get useful analogues in databases" replaced with "Additionally, we did not find useful analogues in databases"
- "Herein, we propose the modification of the isopropyl by intruding double bonds (aldehyde, nitro etc.) can decrease the excitation energy" replaced with "Herein, we propose that modifying the isopropyl by introducing double bonds (e.g., aldehyde, nitro, etc.) can decrease the excitation energy"
- "the molecule with the SMILES of CC1NC(C)=C=C1" should be "the molecule with the SMILES CC1NC(C)=C=C1"; "lies kcal/mol higher than the isomer of in Gibbs energy" should be "lies kcal/mol higher than the isomer in Gibbs energy"
- "with the SMILES of C=CC=NSN=C" should be "with the SMILES C=CC=NSN=C"
- "establish three structural judgment criteria for obtaining molecules with longer excitation wavelengths" replaced with "establish three structural criteria for identifying molecules with longer excitation wavelengths"
- "binary signal dataset labeled based on human experience" should be "a binary signal dataset labeled based on human experience"
- "What we discovered in our application was still consistent." replace with "Our findings in our application were consistent with this."
- "where we updated the parameters of the adapter within the RLHF framework" should be rephrased for clarity: "in which we updated the parameters of the adapter within the RLHF framework."
- "Additionally, we fine-tuned the GPT-2-based model using KTO, DPO, and CPO trainers respectively." should be: "Additionally, we fine-tuned the GPT-2-based model using KTO, DPO, and CPO trainers, respectively."
- "light-responsive molecules in drug delivery systems" should be "light-responsive molecules for drug delivery systems"
- "This diversity in molecule generation tools and content enhances the applicability of these technologies." replaced with "The diversity of molecule generation tools and content enhances the applicability of these technologies."
- "Reinforcement Learning from Human Feedback (RLHF) methods provide an exciting opportunity to incorporate more theoretical chemical knowledge into the generation of high-quality molecular content." replaced with "Reinforcement Learning from Human Feedback (RLHF) methods provide an exciting opportunity to incorporate more theoretical chemical knowledge into generating high-quality molecular content."
- "designing the entire algorithmic framework of the generative pre-trained transformer (GPT) from scratch" replaced with "designing the entire algorithmic framework of the Generative Pre-trained Transformer (GPT) from scratch."

Reviewer 3 Report
Comments and Suggestions for Authors
The manuscript provides a great opportunity to the use of GPT in selection of photo responsive molecules. The major problem is the English writing. The manuscript requires extensive review of the English to help the reader follow the author thought sequence. The method section needs expansion and better explanation to the use of prediction model for excitation energy, drug likeness and evaluation metrics, DFT and TDDFT simulation and RLHF. Figure 1 is very condensed. Please simplify figure 1 into several figures. Figure 2 is not readable and doesn't provide any useful information. A better explanation of figure 2 is needed. Figure 5 is also challenging to understand.
Comments on the Quality of English LanguageSupport in the English is needed to improve the quality of the manuscript
Author Response
Comments: The manuscript provides a great opportunity to the use of GPT in selection of photo responsive molecules. The major problem is the English writing. The manuscript requires extensive review of the English to help the reader follow the author thought sequence.
Reply: Many thanks to the reviewers for pointing out these issues. We have made extensive changes to the language, which will be discussed in detail in the 'Comments on the Quality of English Language' section. These changes have been highlighted as required.
Comments: The method section needs expansion and better explanation to the use of prediction model for excitation energy, drug likeness and evaluation metrics, DFT and TDDFT simulation and RLHF.
Reply: Based on the reviewers' comments, we have added extensive descriptions of RLHF in the Methods section. The specific content is as follows. In addition, we have added details of computations, such as predictive modeling, drug-likeness, and SA, to enhance the annotations of subsequent images.
‘’’
The default sigmoid loss was used in DPO, where the beta factor was set at 0.1. Similarly, the loss type of CPO was sigmoid, and its beta factor was set at 0.1. The beta factor in KTO loss was set at 0.1, with a higher value meaning less divergence from the initial policy. The desirable losses and undesirable losses of KTO are weighed by desirable weight and undesirable weight, respectively. Both of them were set at 1.0.
The preference dataset, used in DPO and CPO, is a dictionary object with the keys of 'prompt,' 'chosen,' and 'rejected.' The binary signal dataset, used in KTO, is a dictionary object with the keys of 'prompt,' 'completion,' and 'label.'
‘’’
.
Comments: Figure 1 is very condensed. Please simplify figure 1 into several figures.
Reply: Based on the reviewers' suggestions, we have added the following to increase the readability of the figure 1.
’’’
Molecules excited by ultraviolet light have the potential to become stimuli-responsive materials in drug delivery systems. The dark green section provides information about the pre-trained transformer. The indigo section shows the pre-training of GPT-2 combined with the SMILES-Tokenizer on the PubChem dataset. The orange-yellow section indicates that a new adapter was fine-tuned using the ultraviolet light-excited molecules dataset on the pre-trained GPT-2. The orange-red section shows the prediction model, trained with the QM7b dataset and Coulomb matrix features, predicting the properties of molecules generated by the fine-tuned GPT-2.
’’’
Comments: Figure 2 is not readable and doesn't provide any useful information. A better explanation of figure 2 is needed.
Reply: Based on the reviewers' suggestions, we have added the following to increase the readability of the figure 2.
‘’’
These molecules were generated by GPT-2 fine-tuned on the ultraviolet light-excited molecules dataset. The excitation energy data in (a) comes from predictions given by an SVM model. (b) shows the distribution of the drug-likeness scores of the molecules, obtained from DeepChem's calculation of the quantitative estimate of drug-likeness (QED) values. (c) presents the synthetic accessibility scores of the molecules, calculated using RDKit, with higher values indicating that the molecules are easier to synthesize.
’’’
Comments: Figure 5 is also challenging to understand.
Reply: Based on the reviewers' suggestions, we have added the following to increase the readability of the figure 5.
‘’’
In the training and fine-tuning of the GPT-2 Peft model, the parameters in the orange-yellow section were used for training, while the parameters of the other layers retained their pre-trained values. The KTO algorithm used Binary Signal Datasets; both DPO and CPO used Preference Datasets. Refer to the Method Section for descriptions of the Binary Signal Datasets and Preference Datasets. Labels in the Binary Signal Datasets are directly assigned by experts to the molecules, with labels of either True or False. In the Preference Datasets, experts select molecules as recommended molecules and identify unreasonable molecules as restricted molecules.
’’’
Comments on the Quality of English Language
Support in the English is needed to improve the quality of the manuscript
Reply: Based on the reviewers' comments and our rechecking, we have made the following changes. All changes have been highlighted as required. In this round of responses to the reviewers' comments, due to time constraints and other reasons, we have temporarily chosen to make the language modifications ourselves. If this still does not meet the publication requirements of the journal, we will contact MDPI's academic editors in the next round of reviews.
- "the Generative Pre-trained Transformer (GPT) has been employed for generating molecular structures" (Introduction): " for generating " replaced with "to generate"
- "these evidences indicate that our application of the combined GPT and TDDFT calculations in designing stimulus-responsive molecules holds practical value for drug delivery" (Introduction): "evidences" replaced with “evidence”
- "the forecasting of temporal dynamics in drug delivery has been accomplished through the application of convolutional neural networks and long short-term memory networks [19]" (Introduction): "forecasting" replaced with "prediction".
- "benefiting from the open-source QM7b datasets provided by Quantum-Machine Project [34,35], our photoresponsive molecular design for drug delivery has gained important datasets [36,37]" (Results and Discussion): replaced with “The open-source dataset QM7b, provided by Quantum-Machine Project, encompasses a variety of physicochemical properties of molecules. In order to ultimately achieve the generation of light-responsive drug delivery molecules, this data was used to fine-tune the pre-trained language model”.
- "to date, studies haven't shown a clear path for applying GPT technology to drug delivery molecules and validating its efficacy" (Results and Discussion): "clear path" replaced with "established methodology"
- "The linked characters of a molecular SMILES give out the information about the chemical bond and molecular configuration" (Results and Discussion): "give" replaced with “gives”
- "The SMILES-Tokenizer is also deal with" replaced with "The SMILES-Tokenizer also deals with."
- "Once this workflow is proved to work" replaced with "Once this workflow is proven to work."
- "This workflow to generate stimulus-responsive molecules via pre-training language model could be found in Fig 1." replaced with "This workflow for generating stimulus-responsive molecules via a pre-trained language model can be found in Fig 1."
- "utilized as input" replaced with be "which are utilized as input."
- "crucial for our adapter training and serving as the foundation of our workflow" replaced with "which are crucial for our adapter training and serve as the foundation of our workflow."
- ”Quantitative structure–activity relationship (QSAR) of molecules stems from the direct influence of chemical bonding, atomic potentials, and molecular conformation on their properties.” replaced with " Understanding the differences in the properties of various molecules based on chemical bonds, atomic potentials, and molecular conformations is a direct manifestation of the quantitative structure-activity relationship (QSAR) of molecules."
- "RDkit" replaced with "RDKit"
- "Higher QED values suggest increased likelihood of drug-likeness" replaced with "Higher QED values suggest an increased likelihood of drug-likeness."; "Synthetic Accessibility Score calculation methods aimed" replaced with "the Synthetic Accessibility Score calculation methods aimed"
- "rbf, sigmoid and quantum" replaced with "rbf, sigmoid, and quantum"; "regularization parameter (from 0.01 to 80)" replaced with "the regularization parameter (from 0.01 to 80)"; "kernel coefficient (auto, scale, 0.8, 0.84, and 2.3)" replaced with "the kernel coefficient (auto, scale, 0.8, 0.84, and 2.3)"; "Before we used our SVR of excitation energies to predict the value of generative molecules, the deepchem was used to calculate the Coulomb matrix of generative modelcules." has been removed; "the deepchem" replaced with "DeepChem"; "The results could be found in Figure 2a" replaced with "The results can be found in Figure 2a"
- "The parameter-tuned UVGPT inherited the pre-trained model's performance on PubChem, with the distribution calculated using the QED method illustrated in Figure 2b." replaced with "The parameter-tuned UVGPT inherited the pre-trained model's performance on PubChem, and the distribution calculated using the QED method is illustrated in Figure 2b."
- "molecules with SMILES of OC1CC1OC(C)C" replaced with "molecules with the SMILES OC1CC1OC(C)C"
- "Both of the above further contribute to the evidence supporting the effectiveness of UVGPT." replaced with "Both findings further contribute to the evidence supporting the effectiveness of UVGPT."
- "the molecule with the SMILES representation of C=CC=NSN=C" replaced with "the molecule with the SMILES representation C=CC=NSN=C"
- "Additionally, the molecule with the SMILES representation of CC(C(Cl)N)S=N" replaced with "Additionally, the molecule with the SMILES representation CC(C(Cl)N)S=N"; "classifying it as cumulene" replaced with "classifying it as a cumulene."
- "These have been shown in Figure 4" replaced with "These are shown in Figure 4."
- "excitation energy properties" replaced with "excitation energies"
- "the vertical excitation by DFT calculations were performed" replaced with "vertical excitations by DFT calculations were performed"
- "generated molecules contains heteroatom" replaced with "generated molecules contain heteroatoms; "Sulfer" replaced with "sulfur"; "atoms suggesting the prospects for biological applications" replaced with "atoms, suggesting prospects for biological applications"
- "The remaining 5 molecules" replaced with "The remaining five molecules"
- "the molecule, with the SMILES of OC1CC1OC(C)C" replaced with "the molecule with the SMILES OC1CC1OC(C)C"
- "And we did not get useful analogues in databases" replaced with "Additionally, we did not find useful analogues in databases"
- "Herein, we propose the modification of the isopropyl by intruding double bonds (aldehyde, nitro etc.) can decrease the excitation energy" replaced with "Herein, we propose that modifying the isopropyl by introducing double bonds (e.g., aldehyde, nitro, etc.) can decrease the excitation energy"
- "the molecule with the SMILES of CC1NC(C)=C=C1" should be "the molecule with the SMILES CC1NC(C)=C=C1"; "lies kcal/mol higher than the isomer of in Gibbs energy" should be "lies kcal/mol higher than the isomer in Gibbs energy"
- "with the SMILES of C=CC=NSN=C" should be "with the SMILES C=CC=NSN=C"
- "establish three structural judgment criteria for obtaining molecules with longer excitation wavelengths" replaced with "establish three structural criteria for identifying molecules with longer excitation wavelengths"
- "binary signal dataset labeled based on human experience" should be "a binary signal dataset labeled based on human experience"
- "What we discovered in our application was still consistent." replace with "Our findings in our application were consistent with this."
- "where we updated the parameters of the adapter within the RLHF framework" should be rephrased for clarity: "in which we updated the parameters of the adapter within the RLHF framework."
- "Additionally, we fine-tuned the GPT-2-based model using KTO, DPO, and CPO trainers respectively." should be: "Additionally, we fine-tuned the GPT-2-based model using KTO, DPO, and CPO trainers, respectively."
- "light-responsive molecules in drug delivery systems" should be "light-responsive molecules for drug delivery systems"
- "This diversity in molecule generation tools and content enhances the applicability of these technologies." replaced with "The diversity of molecule generation tools and content enhances the applicability of these technologies."
- "Reinforcement Learning from Human Feedback (RLHF) methods provide an exciting opportunity to incorporate more theoretical chemical knowledge into the generation of high-quality molecular content." replaced with "Reinforcement Learning from Human Feedback (RLHF) methods provide an exciting opportunity to incorporate more theoretical chemical knowledge into generating high-quality molecular content."
- "designing the entire algorithmic framework of the generative pre-trained transformer (GPT) from scratch" replaced with "designing the entire algorithmic framework of the Generative Pre-trained Transformer (GPT) from scratch."

Round 2
Reviewer 1 Report
Comments and Suggestions for Authors
Based on the provided comments and responses, the authors have addressed the concerns raised by the reviewer. Here’s a detailed evaluation of each comment and the corresponding reply:
1/Reviewer’s concern: the model does not fully integrate extensive theoretical chemical knowledge, which is crucial for molecular design.
Author’s reply: the authors acknowledge this limitation and elaborate on their approach, explaining how they incorporated Density Functional Theory (DFT) to address these gaps. They also discuss the potential of advanced large language models (LLMs) to handle more complex chemical knowledge in the future.
Evaluation: the authors provide a thorough explanation of their current limitations and future directions. They address the concern by outlining their efforts to incorporate more advanced techniques and acknowledge the need for further improvements.
2/reliance on the QM7b dataset:
Reviewer’s concern: QM7b dataset may not fully represent the diversity of molecules relevant to drug delivery, suggesting that a more specialized dataset could improve the model’s performance.
Author’s reply: the authors agree with the reviewer’s point and explain their current reliance on QM7b due to the lack of specialized data. They outline their future plans to expand the dataset by incorporating TDDFT calculations and other sources to improve relevance.
Evaluation: the authors’ response is satisfactory as they acknowledge the limitation and present a clear plan for future work to address this issue.
3/ Validation of generated molecules:
Reviewer’s concern: validation was based on computational simulations, and experimental validation would strengthen the study.
Author’s reply: the authors agree and explain the challenges associated with synthesizing and experimentally validating the molecules. They propose enriching their model with more comprehensive chemical knowledge before proceeding to experimental validation.
Evaluation: the authors provide a reasonable justification for their current approach and outline a logical next step. Their plan to gather more data before experimental validation is practical and aligns with the reviewers' suggestions.
Therefore, based on the provided responses and the revised manuscript, it is reasonable to conclude that the manuscript has sufficiently addressed the reviewers' concerns and is acceptable for publication.
Reviewer 3 Report
Comments and Suggestions for Authors
It is not clear to me if the authors have done any modifications or additions to the manuscript. The yellow highlighted lines are similar in the original and the revised manuscript.
Comments on the Quality of English LanguageEnglish needs some review